# Calibration of Quartz-Enhanced Photoacoustic Sensors for Real-Life Adaptation

**DOI:** 10.3390/molecules26030609

**Published:** 2021-01-25

**Authors:** Jesper B. Christensen, David Balslev-Harder, Lars Nielsen, Jan C. Petersen, Mikael Lassen

**Affiliations:** Danish Fundamental Metrology, Kogle Allé 5, 2970 Hørsholm, Denmark; jbc@dfm.dk (J.B.C.); dbh@dfm.dk (D.B.-H.); ln@dfm.dk (L.N.); jcp@dfm.dk (J.C.P.)

**Keywords:** photoacoustics, gas spectroscopy, environmental sensors, carbon dioxide, humidity, optics, lasers, metrology, calibration

## Abstract

We report on the use of quartz-enhanced photoacoustic spectroscopy for continuous carbon-dioxide measurements in humid air over a period of six days. The presence of water molecules alters the relaxation rate of the target molecules and thus the amplitude of the photoacoustic signal. Prior to the measurements, the photoacoustic sensor system was pre-calibrated using CO_2_ mole fractions in the range of 0–10^−3^ (0–1000 ppm) and at different relative humidities between 0% and 45%, while assuming a model hypothesis that allowed the photoacoustic signal to be perturbed linearly by H_2_O content. This calibration technique was compared against an alternative learning-based method, where sensor data from the first two days of the six-day period were used for self-calibration. A commercial non-dispersive infrared sensor was used as a CO_2_ reference sensor and provided the benchmark for the two calibration procedures. In our case, the self-calibrated method proved to be both more accurate and precise.

## 1. Introduction

The photoacoustic spectroscopy (PAS) technique is finding increasing interest as a powerful, yet simple, trace-gas detection method [1,2,3,4,5,6,7]. PAS, however, is not an absolute metrological technique, and its use requires detailed knowledge of the chemical composition of the investigated gas sample. This renders photoacoustic (PA) sensors difficult to calibrate, and therefore commercialize, for environmental monitoring applications [8].

The PAS method is different from other optical absorption-based methods, most notably the tunable diode laser absorption spectroscopy (TDLAS), cavity ring-down spectroscopy, and non-dispersive infrared absorption spectroscopy (NDIR) [9,10,11,12,13,14,15]. Whereas these methods directly measure optical attenuation, PAS is based on the detection of acoustic waves generated by exciting ro-vibrational states of target molecules with modulated light [16,17,18]. The absorbed energy translates into kinetic energy, which forms an acoustic wave that can be detected with a pressure transducer. In a simplistic view, the generated PA sound-wave intensity depends linearly (in the low-absorption regime) on the concentration of the target molecule and a spectral overlap between the molecular absorption cross section and the light spectrum. However, from a more general perspective, the generated PA sound-wave intensity depends not only on the concentration of the target molecule, but on the entire gas-sample composition. Other molecules may alter the relaxation kinetics of the various excited ro-vibrational modes [19,20], causing a change in the PA signal strength. In this way, the measured PA signal is gas-matrix dependent, meaning that PA-based trace-gas sensors, while they can be extremely sensitive, can quickly become inaccurate without adequate calibration of the necessary gas-matrix corrections. Most notably, and highly relevant for real-life adaptation of PA sensors, the presence of water vapor in a gas sample acts as an enzyme of the relaxation process, and thereby enhances, or in fewer cases diminishes, the generated sound wave. In practice, this entails that absolute environmental gas-concentration measurements can only be achieved upon correction of the sample water content.

Multiple strategies and methods have so far been implemented for humidity correction of PA trace-gas sensors [21,22,23,24,25,26,27,28,29]. Here, we develop a calibration framework based on a simple learning-based method for quantifying the influence of humidity on photoacoustic carbon-dioxide concentration measurements. Using this approach, the model is only required to be locally accurate (within the observed values), which is a highly relaxed assumption. We compare the long-term performance of a commercial NDIR CO${}_{2}$ sensor with that of a quartz-enhanced photoacoustic (QEPAS) module (for details on the QEPAS technique, see, e.g., [30,31,32]) resonantly pumped by a pulsed optical parametric oscillator (OPO) having an emission wavelength of 4.32 μm. We find very good agreement with the NDIR sensor when calibrated using atmospheric measurement data as training data for the calibration algorithm.

## 2. Experimental Setup

The experimental setup, sketched in Figure 1, includes a mid-infrared (MIR) pulsed optical parametric oscillator (OPO), a QEPAS sensor module (ADM01, Thorlabs, Dachau, Germany), an NDIR sensor (T6613, Telaire, Telaire Pforzheim, Germany), optical detectors for power measurement, humidity/temperature/pressure sensors, a mass-flow control system, and a Raspberry PI microprocessor for data acquisition.

The light source is based on an actively Q-switched nanosecond Nd:YAG pump laser (BrightSolution, Anchorage, AK, USA), which emits pulses with a duration of 15 ns at a repetition rate of 12.457 kHz and a center wavelength of 1064 nm. The near-infrared pulses are focused into a 40 mm long fan-out structured periodically poled lithium niobate crystal (HC Photonics, HsinChu City, Taiwan) placed inside a 55 mm long linear cavity with a waist of approximately 150 μm at the cavity midpoint. The two cavity mirrors are characterized by having radius-of-curvature of 100 mm and a high reflectance (R>0.99) in the spectral region from 1350 nm to 1700 nm. Exploiting the resonant signal enhancement in this wavelength range, a continuous chirp of the nonlinear crystal facilitates efficient generation of MIR pulsed light from 2.8 μm to 4.5 μm. In this work, the MIR wavelength is fixed to 4.32 μm and with 15 mW of mean optical output power, matching ro-vibrational lines in the P-branch of the asymmetric stretchband of CO${}_{2}$. More details on the MIR OPO can be found in [33].

The QEPAS module contains a quartz tuning fork (QTF) with an eigenfrequency of $f_{0}$ = 12,457 Hz and a quality factor of ∼5300±50 at 1 atm [34]. The QTF is piezo-electrically active in the mechanical mode for which the two prongs oscillate 180 degrees out of phase (asymmetrical stretching mode), and is therefore sensitive to a pressure wave originating from in between the two prongs and less sensitive to external sound waves which makes the two prongs oscillate in-phase [30,31]. Acoustic coupling is further improved by two microresonator tubes each having a length of 12.4 mm [1]. In- and outcoupling of the mid-infrared light through the module happens through two BaF${}_{2}$ windows with a combined transmittance of ∼0.9. QEPAS is a powerful technique and has shown the capability to monitor gas concentrations at the part per billion or even part per trillion levels [30,31,35,36,37,38].

The gas-flow control is realized using a triplet of Brooks 0254 mass-flow controllers (MFCs). Two MFCs are used for setting the in-flow rate of N${}_{2}$ and 1000-ppm CO${}_{2}$ in an N${}_{2}$ matrix. The N${}_{2}$-CO${}_{2}$ gas flow is combined with a valve-controlled inlet that enables suction of laboratory air into the tube system using a mini vacuum pump with variable flow rate. The combined gas flow is led through the third MFC, which is used to monitor and log the total gas flow, and on to the QEPAS module and the NDIR sensor.

Data processing is enabled by two lock-in amplifiers receiving the electrical local oscillator signal from the active laser Q-switch and with integration times of 300 μs. The first lock-in amplifier measures the incident optical power just before the QEPAS module, and the second lock-in amplifier demodulates the PA output signal of the QEPAS’ in-build transimpedance amplifier using a 1-f configuration (i.e., amplitude modulation) [39,40]. The output from the lock-in amplifiers are digitized using a 10-bit ADC (MCP3008) and logged using a Raspberry PI 3 module (RPI). The same RPI simultaneously logs data from the NDIR sensor and a humidity–temperature–pressure sensor (BME280, Adafruit, New York, NY, USA).

## 3. Theory

A PA sensor can in the simple case be mathematically modeled by assuming a PA signal that depends linearly on both the target-molecule concentration and the optical power used for probing the gas sample. However, for gas samples with a non-zero water-vapor concentration of $c_{H_{2}O}$, the PA signal can be significantly altered. Considering the PA measurement of CO${}_{2}$ concentration, the PA voltage signal, $U_{\mathrm{PA}}$, may therefore be described by a relation of the form
(1)$$U_{\mathrm{PA}}=\alpha_{1}c_{{\mathrm{CO}}_{2}}\left(1+{\mathrm{Poly}}_{n>0}\left(c_{H_{2}O}\right)\right)P,$$
wherein *P* is the optical power, $\alpha_{1}$ is related to the optical absorption of CO${}_{2}$ at the employed wavelength, and ${\mathrm{Poly}}_{n>0}\left(c_{H_{2}O}\right)$ is a constant-excluding polynomial with respect to $c_{H_{2}O}$. To test and compare our calibration method with related published work, the PA signal is hypothesized to be perturbed linearly by water content [24,25], i.e., ${\mathrm{Poly}}_{n}\left(c_{H_{2}O}\right)=\alpha_{\mathrm{AH}}c_{H_{2}O}$. This reforms Equation (Equation 1) to
(2)$${\bar{U}}_{\mathrm{PA}}=\alpha_{1}c_{{\mathrm{CO}}_{2}}\left(1+\alpha_{\mathrm{AH}}c_{H_{2}O}\right).$$

The power-normalized PA signal is defined as ${\bar{U}}_{\mathrm{PA}}\equiv U_{\mathrm{PA}}/U_{P}$, whereof $U_{P}$ is the power-logged voltage signal. Using $U_{P}$, rather than the actual optical power, results in an arbitrary scaling of $\alpha_{1}$ of $\alpha_{1}\to \alpha_{P}\alpha_{1}$, where $\alpha_{P}=\left(\partial P/\partial U_{P}\right)$ is the power-to-voltage linear coefficient of the power meter.

As outlined in Section 2, our experimental setup contains an NDIR CO${}_{2}$-sensor placed in series with the PA module. The NDIR module is based on differential optical absorption at λ≈4.3 μm, and is practically unaffected by water concentration [15]. The NDIR output is highly linear with respect to CO${}_{2}$ concentration, i.e., $c_{{\mathrm{CO}}_{2}}$ ∼ $a_{\mathrm{NDIR}}U_{\mathrm{NDIR}}+b_{\mathrm{NDIR}}$, as is confirmed below. This allows the NDIR module to function as a CO${}_{2}$ reference. Insertion of the linear NDIR response into Equation (Equation 2) yields the relationship between PA output and NDIR output, given as
(3)$${\bar{U}}_{\mathrm{PA}}=a_{0}+a_{1}U_{\mathrm{NDIR}}+a_{2}U_{\mathrm{NDIR}}c_{H_{2}O}+a_{3}c_{H_{2}O},$$
where the coefficients $a_{0}=b_{\mathrm{NDIR}}\alpha_{1}$, $a_{1}=a_{\mathrm{NDIR}}\alpha_{1}$, $a_{2}=a_{\mathrm{NDIR}}\alpha_{1}\alpha_{\mathrm{AH}}$, and $a_{3}=b_{\mathrm{NDIR}}\alpha_{1}\alpha_{\mathrm{AH}}$.

Equation (Equation 3) provides a model function with which the PA sensor can be calibrated with reference to the NDIR sensor for varying carbon-dioxide and water concentrations. The water-vapor concentration is computed based on the BME280 output (relative humidity, pressure, and temperature), and is expressed as [41]
(4)$$c_{H_{2}O}=\frac{Px_{H_{2}O}}{RT}=\frac{611.6\;\mathrm{Pa}}{RT}\exp\;\left[\frac{17.48\left(T-273.15\;K\right)}{T-32.42\;K}\right]\mathrm{RH},$$
where $x_{H_{2}O}$ is the molar fraction of water vapor, $R\equiv 8.314462618\;J\;{\mathrm{mol}}^{-1}\;K^{-1}$, RH is the relative humidity expressed as a fractional number between 0 and 1, *T* is the absolute temperature measured in kelvin, and *P* is the pressure in pascal. The calculated water-vapor concentration is associated with a standard uncertainty calculated using the laws of error propagation while using the rated uncertainties of the sensor of u(T)=1K, u(RH)=3%, and u(P)=100Pa.

## 4. Experiments

### 4.1. Calibration of Carbon-Dioxide Sensors (Dry)

We first assess the CO${}_{2}$-response of the two sensors in dry conditions, i.e., $x_{H_{2}O}=0$. To this end, the lab-air valve is closed off, and the MFCs are used to achieve different CO${}_{2}$ concentrations (in $N_{2}$). The CO${}_{2}$ concentration is varied from 0 to $10^{-3}$ in steps of ∼125 ppm while preserving a constant flow rate of 80 mL/min. For each CO${}_{2}$-level, measurements are conducted for 200 s resulting in 100 and 50 measurement points for the QEPAS and NDIR sensors, respectively. With the expectancy of a linear relationship between CO${}_{2}$ concentration and output signal, a linear regression is performed on the flow- and power-normalized data, see Figure 2. The NDIR sensor calibration curve Figure 2a, which displays linear behaviour over the entire measurement range, is given by $U_{\mathrm{NDIR}}=\beta_{\mathrm{NDIR}}+\alpha_{\mathrm{NDIR}}c_{{\mathrm{CO}}_{2}}=440.73\;\mathrm{mV}+0.0324\;\mathrm{mV}\;m^{3}\;{\mathrm{mol}}^{-1}c_{{\mathrm{CO}}_{2}}$ with parameter uncertainties of $u\left(\alpha_{\mathrm{NDIR}}\right)=5\times 10^{-5}\;\mathrm{mV}\;m^{3}\;{\mathrm{mol}}^{-1}$, $u(\beta_{\mathrm{NDIR}})=1.09\;\mathrm{mV}$ and a correlation of $r(\alpha_{\mathrm{NDIR}},\beta_{\mathrm{NDIR}})=-0.85$. The QEPAS sensor Figure 2b is seen to be affected by saturation effects at high CO${}_{2}$ concentration [42]. As a result, only points for which $x_{{\mathrm{CO}}_{2}}<600\;\mathrm{ppm}$ are included in the linear regression analysis. The resulting fit yields ${\bar{U}}_{\mathrm{PA}}=\beta_{\mathrm{PA}}+\alpha_{\mathrm{PA}}c_{{\mathrm{CO}}_{2}}=0.0042+1.2\times 10^{-5}\;m^{3}\;{\mathrm{mol}}^{-1}c_{{\mathrm{CO}}_{2}}$ with parameter standard uncertainties of $u\left(\alpha_{\mathrm{PA}}\right)=2\times 10^{-8}\;m^{3}\;{\mathrm{mol}}^{-1}$, $u(\beta_{\mathrm{PA}})=0.0003$ and correlation $r(\alpha_{\mathrm{PA}},\beta_{\mathrm{PA}})=-0.83$.

### 4.2. Calibration of Carbon-Dioxide Sensors (Wet)

With the purpose of using the PA sensor for atmospheric CO${}_{2}$ monitoring, it is now calibrated to correct for humidity content in the sample gas. This is achieved by using the water-independent, and highly linear, response of the NDIR sensor, as a CO${}_{2}$ reference. Two approaches for calibration are attempted and assessed: (i) the PA response is measured for a range of CO${}_{2}$ and humidity levels and the data are fitted to Equation (Equation 3) using appropriate constraints, and (ii) the natural variations in atmospheric CO${}_{2}$ and water content over a period of three days are used as a basis for establishing a relationship between PA signal, humidity level, and CO${}_{2}$ concentration, in a learning-like fashion (see Section 4.3).

Our statistical analysis is based on a least-squares approach, described in [43], which is in compliance with the “Guide to the Expression of Uncertainty in Measurement” (GUM) [44]. In this framework, each measurand (${\bar{U}}_{\mathrm{PA}}^{\left(i\right)},U_{\mathrm{NDIR}}^{\left(i\right)},c_{H_{2}O}^{\left(i\right)}$) is considered a statistical estimate (with accompanying uncertainty) of the “true” values ($\zeta_{\mathrm{PA}}^{\left(i\right)},\zeta_{\mathrm{NDIR}}^{\left(i\right)},\zeta_{H_{2}O}^{\left(i\right)}$). Each set of measurands are linked through implicit constraints, which, according to Equation (Equation 3), are given by
(5)$$a_{0}+a_{1}\zeta_{\mathrm{NDIR}}^{\left(i\right)}+a_{2}\zeta_{\mathrm{NDIR}}^{\left(i\right)}\zeta_{H_{2}O}^{\left(i\right)}+a_{3}\zeta_{H_{2}O}^{\left(i\right)}-\zeta_{\mathrm{PA}}^{\left(i\right)}=0,\;\forall i.$$

The general least-squares algorithm provides values for the model parameters, $a_{0-3}$, refined estimates of the measurands ($\zeta_{\mathrm{PA}}^{\left(i\right)},\zeta_{\mathrm{NDIR}}^{\left(i\right)},\zeta_{H_{2}O}^{\left(i\right)}$), and the covariance matrix of all determined quantities.

Figure 3a demonstrates how the presence of water molecules enhances the PA signal for a specific CO${}_{2}$-level (quantified by the NDIR sensor response). The data are constituted by four different data series differentiated by the approximate mole fraction of water of 0 (blue), $2\times 10^{-3}$ (green), $4\times 10^{-3}$ (orange), and atmospheric levels of $10\times 10^{-3}$ to $12\;\times 10^{-3}$ (red). Linear trendlines are added to the plot for the three dataseries (with sub-atmospheric humidity levels) to help visualization of the slightly increased slope as a function of humidity level, and the error bars signify standard uncertainties found through repeated measurements.

Running the general least-squares algorithm using the data shown in Figure 3a, provides parameter estimates of $a_{0},a_{1},a_{2}$ and $a_{3}$ with which the model residuals (${\bar{U}}_{\mathrm{PA}}^{\left(i\right)}-\zeta_{\mathrm{PA}}^{\left(i\right)})$ provides a basis for a validity check using a $\chi^{2}$-test. Figure 3b shows the normalized deviations (residuals) between the PA measurands ${\bar{U}}_{\mathrm{PA}}^{\left(i\right)}$ and the refined estimates $\zeta_{\mathrm{PA}}^{\left(i\right)}$ defined as [43]
(6)$$d_{i}=\frac{{\bar{U}}_{\mathrm{PA}}^{\left(i\right)}-\zeta_{\mathrm{PA}}^{\left(i\right)}}{u({\bar{U}}_{\mathrm{PA}}^{\left(i\right)}-\zeta_{\mathrm{PA}}^{\left(i\right)})},$$
and further displays the $\chi^{2}$-value obtained in the fit. With a value of $\chi^{2}=40.3>30=\nu$, where ν is the degrees of freedom, $\mathrm{Pr}(\chi^{2}\left(30\right)>40.3)=0.098$, meaning that the model (null) hypothesis is rejected at a 10%-significance level, but can not be discarded at a 5%-significance level. However, the observed $\chi^{2}$-value indicates that the model might be too simple to describe the physics at hand in both dry and wet conditions simultaneously. This conclusion is strengthened by the normalized deviations, a considerable amount of which lie outside (or close to the boundaries) of the range [−2;2].

### 4.3. Prolonged Atmospheric Carbon-Dioxide Measurements

We now assess the PA sensor with respect to stability in performance over prolonged periods of multiple days. For a given set of measurands (${\bar{U}}_{\mathrm{PA}},c_{H_{2}O})$, we predict a CO${}_{2}$-concentration of
(7)$$c_{CO_{2}}=b_{\mathrm{NDIR}}+a_{\mathrm{NDIR}}\frac{{\bar{U}}_{\mathrm{PA}}-\left(a_{0}+a_{3}c_{H_{2}O}\right)}{a_{1}+a_{2}c_{H_{2}O}}.$$

The combined model and measurand uncertainty of this prediction is quantified by the variance
(8)$$u^{2}\left(c_{{\mathrm{CO}}_{2}}\right)=d^{T}\Sigma d,$$
where *d* is a column vector containing the partial derivatives of $c_{{\mathrm{CO}}_{2}}$ with respect to parameters $a_{0-3}$, $b_{\mathrm{NDIR}}$, $a_{\mathrm{NDIR}}$, and the measurands ${\bar{U}}_{\mathrm{PA}}$ and $c_{H_{2}O}$, and Σ is the associated eight-dimensional covariance matrix of those same quantities.

Figure 4 illustrates our 6-day long CO${}_{2}$ measurements using both the NDIR sensor and the estimated concentration levels based on three different methods of calculation. Figure 4b compares the NDIR signal with a CO${}_{2}$-measured PA signal uncorrected for the atmospheric water content visualized in Figure 4a for reference. In this case, the CO${}_{2}$ concentration is calculated based on a “dry” calibration (Figure 2), and thus fails to take into account the enhancing effect of the water molecules. As a result, the deduced CO${}_{2}$ data shows strong correlation to the humidity data, and significantly overestimates the CO${}_{2}$ content of the gas. Conversely, Figure 4c shows the estimated CO${}_{2}$ concentration based on the calibration procedure described in Section 4.2. In this situation, the bias offset of the PA-calculated CO${}_{2}$-level is largely removed; however, the water content is still visually observable, indicating that the humidity is not ideally compensated with the computed calibration parameters. Finally, Figure 4d shows the case where the system is allowed to self-calibrate based on “historical” data. The grey area marks the training period, in which data from the three sensors are used to build a model in the same way as outlined in Section 4.2. Using this latter approach, the model in Equation (Equation 2) is only required to be locally accurate (within the observed values of ${\mathrm{CO}}_{2}$ and $H_{2}O$), which is a highly relaxed assumption. As a result, the PA-calculated CO${}_{2}$ level succeeds in predicting the same CO${}_{2}$ level as the reference NDIR sensor. Different sets of calibration parameters for different levels of humidity could in principle be determined and used to calibrate the PA sensor for all humidity values.

## 5. Discussion

Measurements of gas concentrations using PAS becomes highly nontrivial in wet (water-containing) gas mixtures. Although water molecules do not, necessarily, directly contribute through optical absorption, they influence the relaxation mechanisms of other absorbing molecules in the mixture. This effect is apparent from Figure 4b, in which the PA signal is seen to be enhanced by more than a factor of 1.5 as a result of the humidity levels ($x_{H_{2}O}∼10\times 10^{-3}$) displayed above in Figure 4a. This enhancement has previously been demonstrated to be both gas- and wavelength dependent, but, perhaps more critically, the enhancement factor does not necessarily seem to be a simple linear function of absolute humidity (see fx [28] for a study on CO). This lag of linearity can also be found in our data for CO${}_{2}$, e.g., by comparing data at time stamps t=15h and t=60h, for which we estimate water-enhancement factors, $\alpha_{\mathrm{AH}}$ of $\left(0.0015\pm 0.0002\right)\;m^{3}\;{\mathrm{mol}}^{-1}$ and $\left(0.0008\pm 0.0002\right)\;m^{3}\;{\mathrm{mol}}^{-1}$, respectively. It is this inconsistency that results in the inaccurate correction performed in Figure 4c, and which ultimately means that the model hypothesis must be discarded as being too simple to describe the physics at hand.

To investigate if the established setup could still be used for CO${}_{2}$ monitoring, we attempted to base the calibration on “historical” data from the PA sensor, the humidity sensor, and the NDIR sensor as a CO${}_{2}$ reference. After two days of model training in normal atmospheric conditions, our PA sensor proved to be capable of estimating the same CO${}_{2}$ level (within the uncertainties) as the reference NDIR sensor throughout a period of four days. This also included the final tests of daylong human interference seen towards the end of the considered time span that lasted from the 5th to the 10th of November 2020.

Our analysis here involved the use of a cheap miniature sensor for monitoring the atmospheric humidity. Such sensors are typically fairly inaccurate from an absolute perspective, resulting in the large confidence interval in Figure 4c, and a smaller relative uncertainty leading to the smaller confidence interval in Figure 4d. However, even in the latter case, the humidity uncertainty is still the largest contribution in the final uncertainty budget of the photoacoustically-based CO${}_{2}$ measurements. This strongly underlines that PA-based concentration measurements in real-life monitoring conditions necessarily involve a highly sensitive measurement of the absolute humidity level. A sensitive humidity measurement can either be done using the PA effect [25], or, perhaps more conveniently, by embedding small state-of-the-art humidity sensors (see e.g., [45,46]) into the PA gas chamber.

## 6. Conclusions

In this work, we presented a pilot study of quartz-enhanced photoacoustic carbon-dioxide measurements in air of varying humidity. With our general least-squares statistical framework, we tested a model function for which the water-concentration was assumed to perturb the photoacoustic signal to first order. However, the calibration, and subsequent prolonged carbon-dioxide measurements, resulted in a rejection of the model hypothesis. Instead, the photoacoustic sensor was calibrated using atmospheric measurement data from the first two days, of a six-day period, as training data. Upon this calibration, the photoacoustic sensor was found to provide carbon-dioxide estimates that were in agreement with the reference non-dispersive infrared module over the entire test period.

## Figures and Tables

**Figure 1 molecules-26-00609-f001:**
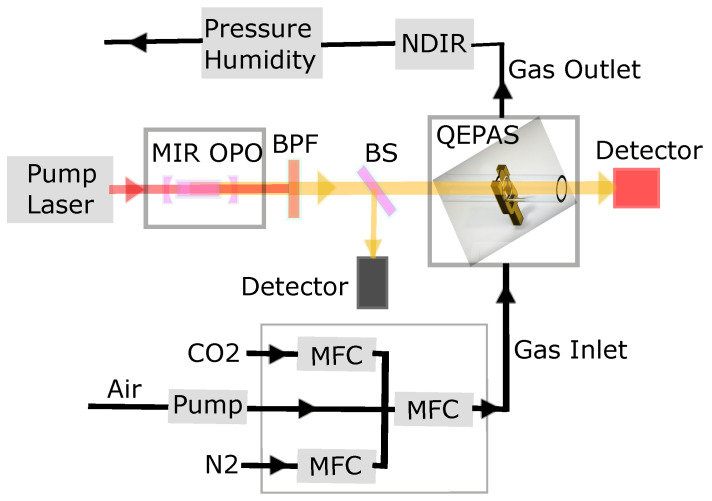
Block diagram of the main parts of the experimental setup. QEPAS: Quartz-enhanced PAS. MIR OPO: Mid-infrared (MIR) pulsed optical parametric oscillator. MFC: Mass-flow controller. BPF: Bandpass filter. BS: Beamsplitter. Two detectors (black: fast detector; red: thermal detector) are monitoring the optical power.

**Figure 2 molecules-26-00609-f002:**
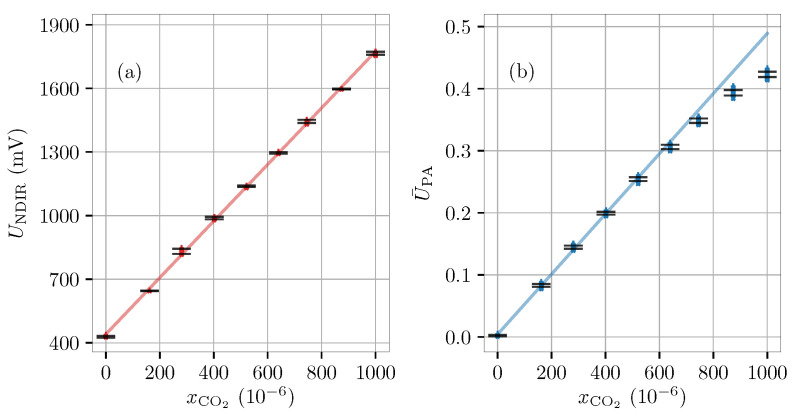
Calibration of carbon-dioxide sensor response in dry nitrogen gas samples. (**a**) NDIR sensor response, and (**b**) power-normalized QEPAS sensor response. The error bars (*y*-direction) identify the 1σ variation of datapoints around the mean value of each dataset. During the calibration procedure, the pressure and temperature were P=(1012±1)hPa and $T=\left(22.8\pm 0.1\right){\;}^{\circ }C$, respectively.

**Figure 3 molecules-26-00609-f003:**
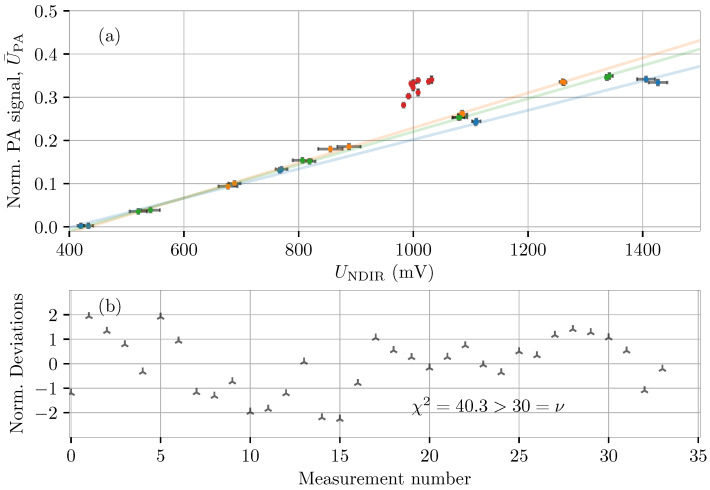
Calibration of a PA sensor of CO${}_{2}$ in humid gas samples. (**a**) acquired PA signal for a wide range of CO${}_{2}$- and absolute humidity levels, demonstrating the PA water-enhancement effect. The error bars represent 1σ standard deviations for each measurement point; (**b**) the normalized deviations, $d_{i}$ for each of the 34 measurand sets, acquired using the generalized least-squares algorithm [43].

**Figure 4 molecules-26-00609-f004:**
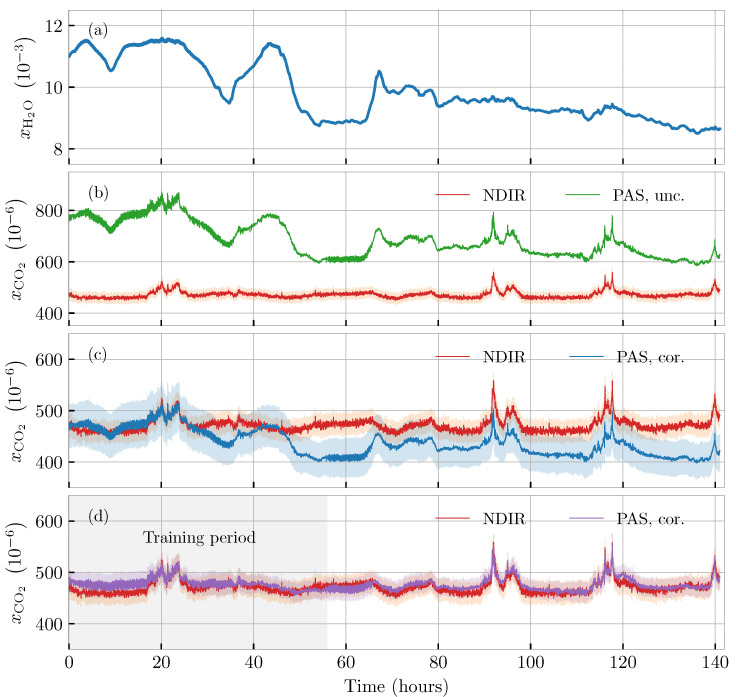
Carbon-dioxide monitoring over six days in atmospheric humidity levels. (**a**) measured absolute humidity; (**b**) CO${}_{2}$ level deduced from PA signal, uncorrected for the atmospheric water content; (**c**) CO${}_{2}$ level deduced from PA signal, corrected for the atmospheric water content using parameters from the calibration (Figure 3); (**d**) CO${}_{2}$ level deduced from PA signal, corrected for the atmospheric water content using a model based on “historical” training data from the shaded time period. The shaded regions around each curve represent the calculated 1σ confidence region for each time series.

## Data Availability

The data presented in this study are available on request from the corresponding author.

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
