# Peer review of "Calibration of Quartz-Enhanced Photoacoustic Sensors for Real-Life Adaptation"

_molecules, 2021, doi:10.3390/molecules26030609_

Round 1
Reviewer 1 Report
This paper report the correction of QEPAS signal in the presence of different humidity. Indeed, water vapor can affect the relaxation rate of some gas molecules such as CO2, which affect the signal amplitude of the photoacoustic spectroscopy. The authors solved the influence of water vapor concentration on the photoacoustic signal through machine learning method. The results shown that the CO2 concentration measured by the QEPAS sensor through machine learning method has very good consistency with the NDIR sensor. This make the MS is interesting for readers and community of spectroscopy gas sensing. The subject of the manuscript falls within the scope of the Molecules journal and it can be accepted for the publication after revisions.
Comments:
- The relative humidity in the atmosphere is usually as high as 90%. Why does the author only use 0%-45% relative humidity when calibrating? Is there a problem with humidifier used?
- There should be given explanation on the different lines in figure 3(a)
(3) The author uses machine learning to improve the accuracy of CO2 concentration measurement by photoacoustic spectroscopy. This is the main innovation of this article, and some descriptions about machine learning should be added.
(4) During the continuous measurement, does the author regularly determine the resonance frequency of the quartz tuning fork? The resonance frequency of the quartz tuning fork drifts with changes in the ambient temperature. If the resonant frequency of the quartz tuning fork is not calibrated regularly, the measured CO2 concentration will also be inconsistent with the NDIR sensor.
(5) A figure of comparison between original QEPAS CO2 signals and corrected signal with different H2O level should be given in the MS.
Author Response
We would like to thank the referee for reading of our paper. We are delighted that our manuscript can be considered for publication. We highly appreciate the detailed and valuable comments of the referees, which have contributed to a significantly improved manuscript. We have addressed the comments below.
- To realize different humidity levels, we mix atmospheric air (of known/measured humidity) with N2 or N2-diluted CO2. As the relative humidity of the ambient air in the lab did not exceed 45%, the calibration range was naturally limited.
- Good point, we write:
“Linear trendlines are added to the plot for the three data series (with sub-atmospheric humidity levels) to help visualization of the slightly increased slope as a function of humidity level, …” - The “machine learning” procedure is not so different from the described regular calibration procedure. However, rather than relying on premeasurements using widely different CO2 and H2O levels, the “learning” procedure uses data from the first two days of measurement on ambient air to calibrate the sensor. In the paper we write:
“and ii the natural variations in atmospheric CO2 and water content over a period of three days are used as a basis for establishing a relationship between PA signal, humidity level, and CO2 concentration, in a learning-like fashion” - The resonance frequency of the QTF and its quality factor were measured regularly to ensure degradation-free operation and to monitor the spectral drift of the resonator. The minimal observed spectral drift was concluded to provide an insignificant error source during the measurement period.
- We do not expect the learning-based model to be accurate at, for example, a RH of 0%, unless a new learning period is applied to the sensor and new parameters are being determined. Using this approach, the sensor is only locally calibrated using the variations of it’s measurement environment. However, different sets of calibrations parameters for different levels of humidity could in principle be determined. So, although such curve could perhaps indicate the validity range of the sensor, we opted not to make it.
Reviewer 2 Report
In this paper, a calibration method for quartz-enhanced photoacoustic spectroscopy was presented. Carbon dioxide was used as the analyte. I think this research is interesting and enriched the photoacoustic spectroscopy community. Therefore, it can be accepted for publication after the following issues are addressed.
- The author introduced QEPAS technique in the Section 2. I recommend to describe it in the Introduction Section. And some latest references can be cited: a) such as “ In plane quartz-enhanced photoacoustic spectroscopy.” Applied Physics Letters. 2020, 116, 061101.
- I am not very sure about the deduction in Section 3. For example, the author stated that “Hypothesizing that the PA signal is perturbed linearly by water content”. What is the basis for “linearly”? Please give the reference. Also please see and cite the following reference to find the similar work for the water content. a) QEPAS based ppb-level detection of CO and N2O using a high power CW DFB-QCL. Optics Express. 2013, 21(1): 1008-1019. b) Sensitive methane detection based on quartz-enhanced photoacoustic spectroscopy with a high-power diode laser and wavelet filtering. Optics and Lasers in Engineering. 2020, 132, 106155.
- In the Experiments Section, the author used PAS. But it should be QEPAS. PAS meaning traditional method using microphone, cantilever and so on. QEPAS uses quartz tuning fork.
Author Response
We would like to thank the referee for reading of our paper. We are delighted that our manuscript can be considered for publication. We highly appreciate the detailed and valuable comments of the referees for making our manuscript significantly better. We have addressed the comments below.
- Our explanation of the QEPAS module is placed in the experimental section, which we think fitting for the current manuscript as it is not crucially important how QEPAS works to a non-expert. Instead we now write in the introduction:
“(for details on the QEPAS technique, see e.g [26-28])”, adding for example the reference suggested by the reviewer.
- We have added the reference that suggests a linear relationship based on molecular relaxation kinetics. Linear perturbations are by far the most common form of perturbations, as also indicated in a number of publications (see [22] and [23]) so it should not as such be necessary to provide a reason for hypothesizing it.
We have also added the two references suggested by the reviewer to our list. - As we see it, QEPAS is a subcategory of PAS. Our title says “Quartz-enhanced photoacoustic sensor”, so we don’t believe anyone will be confused that we call our sensor a PA sensor rather than a QEPAS sensor.
Reviewer 3 Report
The manuscript is well written and the research was conducted properly. My main doubts are about the novelty. The fact that water vapour inteferes with absorption measurements of other gasous species (in particular CO2) is well known and have been studied for decades. Hence, the statement "Our work underpins the
importance of applying corrections to amount of substance photoacoutistc measurements of carbon
dioxide in the presence of varying water-concentration." does not add any new value to the current state of knowledge. Use of corrections based on experimental calibration of sensors is also a well known method. For this reason the sentence "Our work establishes a methodology for performing these corrections." is arguable.
Author Response
We thank the reviewer for her/his assessment of our work.
We would like to point out that our paper is indeed highly novel. It is true that the influence of humidity on the PAS signal has been investigated in several papers (see for example citation 22 and 23), where it was concluded that there is a linear relationship between the PA signal and the humidity. However, as we show, this is only true for small variations in humidity. This means that the linear model hypothesis must be discarded as being too simple to describe the physics at hand.
Based on the criticism, we have changed the above sentences in our abstract to:
“Both a traditional calibration method and a learning-based procedure were used to calibrate the CO2 sensor, the latter proving to be more accurate and precise when compared with the commercial reference sensor.”
Reviewer 4 Report
The authors present a preliminary study of the Relative Humidity as influence parameter to the response of a CO2 sensor based on the photoacoustic response. This reviewer finds the topic quite out of the scope of this journal.
The paper is interesting to the sensors and/or instrumentation scientific community, however major revisions are required in order to increase the scientific soundness of the paper.
The description of the experimental set-up is very clear, but the description of the calibration experiments should be improved.
In equation (2) the author propose a model of the photoacustic response of the sensor depending on the relative humidity (or water concentration). They do not provide a theoretical background for the model in eq. (2) and their measurements confirm that the model is not valid (Figure 4.c and conclusions). I would recommend to remove the analysis assuming this linear response to the RRHH as influence parameter.
It is also quite confusing the data provided of the NDIR reference sensor. It is supposed that this sensor provides the CO2 concentration independently of the RRHH. This NDIR sensor is factory calibrated and they also have the facilities to calibrate it. They could use the output of this sensor and the associated uncertainty to obtain the reference data of the CO2 concentration. If there is a reason to include in the model the independent parameters aNDIR and bNDIR it must be better explained.
The learning from data approach is very interesting. The authors should better describe the method and the output of this approach. It is expected that the analysis tools provide an empirical calibration of the photoacoustic response against RRHH. A detailed analysis of the validity of the calibration model should be provided.
Author Response
We would like to thank the referee for reading our paper. We are delighted that our manuscript can be considered for publication. We highly appreciate the detailed and valuable comments of the referees for making our manuscript significantly better. We have addressed the comments below.
We would like to note that this is a special topic of the journal Molecules, which is about recent trends and progress in the field of PAS sensing, amongst others. Therefore, we believe that our paper is highly appropriate and deals with one of the main difficulties of PAS sensing, namely absolute concentration calibration in real environments.
It is unclear what part of the calibration experiments the reviewer is referring to. We believe that the experimental part is more than adequately described in the current manuscript. The statistical method used for the sensor calibration is only described in sufficient detail that the reader may understand the most important aspects. For a more in depth understanding of the general least squares, the reader should read ref. 41. The CO2 calibration experiments are simple and very standard. It is realized by using different humidity levels by mixing atmospheric air (of known/measured humidity) with N2 or N2-diluted CO2. As the relative humidity of the ambient air in the lab did not exceed 45%, the calibration range was naturally limited to 0-45%.
The reason for making a calibration/validation of the NDIR sensor is simply to ensure correct linear operation of the NDIR sensor and to estimate the uncertainties on the absolute precision. This is not only important for the model, but also used for benchmarking our QEPAS sensor. While we find that the sensor performance is within specs, our method requires the calibration parameters and their uncertainties/covariance. This we obtain by doing our own calibration. The NDIR signal is also used for the “learning-based” calibration, where the curve together with the two other sensors are recorded and used to set the free parameters.
The linear model was chosen in order to test and benchmark our calibration method with related published work on the effect of humidity, see citation 22 and 23. For example in the work by Yin et al. and Elefante et al. they both demonstrate a strong linear effect on the PAS signal as function of humidity. However, we believe that this is probably due to the limited range of humidity used. While in our work we use a calibration range from 0 to 45%, thus in this case the linear model fails to compensate for the humidity and a precis CO2 concentration estimate is made impossible.
We have added the following line in the theory section:
“To test and compare our calibration method with related published work, the PA signal is hypothesized to be perturbed linearly by water content [22,23], i.e. ….”
[22] Yin, X.; Dong, L.; Zheng, H.; Liu, X.; Wu, H.; Yang, Y.; Ma, W.; Zhang, L.; Yin, W.; Xiao, L.; Jia, S.251Impact of humidity on quartz-enhanced photoacoustic spectroscopy based CO detection using a near-IR252telecommunication diode laser. Sensors2016, 16, 162.
[23] Elefante, A.; Menduni, G.; Rossmadl, H.; Mackowiak, V.; Giglio, M.; Sampaolo, A.; Patimisco, P.; Passaro, V.;Spagnolo, V. Environmental Monitoring of Methane with Quartz-Enhanced Photoacoustic Spectroscopy Exploiting an Electronic Hygrometer to Compensate the H2O Influence on the Sensor Signal. Sensors2020,25620, 2935
The “learning-based” procedure is somehow similar to the described regular calibration procedure. However, rather than relying on measurements using widely different CO2 and H2O levels, the “learning” procedure uses data from two days of measurement on ambient air to calibrate the sensor.
In the paper we write:
“and ii the natural variations in atmospheric CO2 and water content over a period of three days are used as a basis for establishing a relationship between PA signal, humidity level, and CO2 concentration, in a learning-like fashion”
We do not expect the learning-based calibration to be accurate at, for example, a RH of 0%, unless a new learning-based period is applied to the QEPAS sensor and new calibration parameters are being determined. Using this approach, the sensor is only locally calibrated using the variations of it’s measurement environment. So, although such curve could perhaps indicate the validity range of the sensor, we opted not to make it. This is further discussed in the discussion section.
We have added the following sentence in line 158:
“However, different sets of calibrations parameters for different levels of humidity could in principle be determined and used to calibrate the sensor for all humidity values.”
Round 2
Reviewer 1 Report
I think the MS can be accepted as present form
Author Response
We thank the reviewer for his/her positive assessment of our work.
Reviewer 3 Report
My previous remarks stand valid. The authors claim that the main novelty is about the fact that the influence of humidity on the PAS signal is not linear, but the paper is not about the influence of humidity on the PAS signal, but about the calibration method. And the calibration method described in the manuscript (no matter whether the relationship is linear or not) is not novel. So, once again: The fact that water vapour inteferes with absorption measurements of other gasous species (in particular CO2) is well known and have been studied for decades. Use of corrections based on experimental calibration of sensors is also a well known method.
Author Response
We thank the reviewer for his/her assessment of our work.
The paper is about calibration techniques for water-enhancement correction, which remains a subject of active research. Some previous studies suggest a linear perturbation from H2O concentration, but consider only a limited concentration range that excludes data points from dry samples. Our work highlights that this is not sufficient (at least for CO2) to conclude a linear relationship as the “standard” calibration fails to correct for the water content (as this varies). Instead the “learning-based” (local) calibration succeeds.
Reviewer 4 Report
I consider the scientific interest of the paper low. The authors present a simple calibration experiment where they take measurement for two days and use the obtained calibration parameters for the following days. This calibration is valid at the specific conditions of the experiment. It would be worth to analyze the resulting calibration parameters they obtain and justify a more general model or compare it with theoretical studies.
Author Response
We thank the reviewer for his/her assessment of our work.
We very much disagrees with the reviewer on the fact that the paper is of low interest it is a subject of active research within PAS sensors. Most studies on the effects and enhancement of the PAS signal due to humidity has only used small changes in humidity (+/- 2%), thus the linear model is only required to be locally accurate. It has previously been demonstrated that the PAS technique depends both on the gas- and wavelength, perhaps more critically, the enhancement factor does not necessarily seem to be a simple linear function of humidity (see fx [26] for a study on CO). Our learning based method is a new type of approach for absolute calibration of PAS sensors. With our learning based method the photoacoustic sensor was found to provide carbon-dioxide estimates that were in agreement with the reference non-dispersive infrared module over the entire test period (the period can be decreased). The method can easily and directly be applied to the PAS sensor also for different gasses mixtures without the need for a more general model, which makes absolute concentration measurements much more convenient for real life applications.